# Citizen science reveals unexpected solute patterns in semiarid river networks

**Erin Fleming Jones** [1] *, **Rebecca J. Frei**[1¤a], **Raymond M. Lee** [1], **Jordan D. Maxwell**[1], **Rhetta Shoemaker**[1], **Andrew P. Follett**[1¤b], **Gabriella M. Lawson**[1], **Madeleine Malmfeldt**[1], **Rachel Watts**[1], **Zachary T. Aanderud**[1], **Carter Allred**[1], **Allison Tuttle Asay**[1], **Madeline Buhman**[1¤c], **Hunter Burbidge**[1], **Amber Call**[1], **Trevor Crandall**[1¤d], **Isabella Errigo** [1], **Natasha A. Griffin**[1], **Neil C. Hansen**[1], **Jansen C. Howe**[1], **Emily L. Meadows**[1], **Elizabeth Kujanpaa**[1], **Leslie Lange**[1], **Monterey L. Nelson**[1], **Adam J. Norris** [1], **Elysse Ostlund**[1], **Nicholas J. Suiter**[1], **Kaylee Tanner**[1], **Joseph Tolworthy**[2], **Maria Camila Vargas**[1], **Benjamin W. Abbott**[1]

**1** Department of Plant and Wildlife Sciences, Brigham Young University, Provo, Utah, United States of America, **2** Department of Geology, Brigham Young University, Provo, Utah, United States of America

¤a Current address: Department of Renewable Resources, University of Alberta, Edmonton, Alberta, Canada
¤b Current address: Yale Law School, Yale University, New Haven, Connecticut, United States of America
¤c Current address: Office of Research and Development, Environmental Protection Agency, Arlington, Virginia, United States of America
¤d Current address: Cimarron Valley Research Station, Oklahoma State University, Perkins, OK, United States of America
* Erinfjones3@gmail.com

**Data Availability Statement:** All data and code used are available at doi.org/10.4211/hs. 85f3584dccb54afba5698ac615ff949a.

## Abstract

Human modification of water and nutrient flows has resulted in widespread degradation of aquatic ecosystems. The resulting global water crisis causes millions of deaths and trillions of USD in economic damages annually. Semiarid regions have been disproportionately affected because of high relative water demand and pollution. Many proven water management strategies are not fully implemented, partially because of a lack of public engagement with freshwater ecosystems. In this context, we organized a large citizen science initiative to quantify nutrient status and cultivate connection in the semiarid watershed of Utah Lake (USA). Working with community members, we collected samples from ~200 locations throughout the 7,640 km$^2$ watershed on a single day in the spring, summer, and fall of 2018. We calculated ecohydrological metrics for nutrients, major ions, and carbon. For most solutes, concentration and leverage (influence on flux) were highest in lowland reaches draining directly to the lake, coincident with urban and agricultural sources. Solute sources were relatively persistent through time for most parameters despite substantial hydrological variation. Carbon, nitrogen, and phosphorus species showed critical source area behavior, with 10–17% of the sites accounting for most of the flux. Unlike temperate watersheds, where spatial variability often decreases with watershed size, longitudinal variability showed an hourglass shape: high variability among headwaters, low variability in mid-order reaches, and high variability in tailwaters. This unexpected pattern was attributable to the distribution of human activity and hydrological complexity associated with return flows, losing river reaches, and diversions in the tailwaters. We conclude that participatory science has great potential to reveal ecohydrological patterns and rehabilitate individual and community

**Funding:** This work was supported by the Utah Division of Wildlife Resources through the Watershed Restoration Initiative program and National Science Foundation (award numbers EAR-2011439 and EAR-2012123) to BWA. Additional support was provided to BWA, ZTA, and NCH by the Sant Family Foundation through the Roger and Victoria Sant Endowment and the Provo River Watershed Council. The funders had no role in study design, data collection and analysis, decision to publish, or preparation of the manuscript.

**Competing interests:** The authors have declared that no competing interests exist.

relationships with local ecosystems. In this way, such projects represent an opportunity to both understand and improve water quality in diverse socioecological contexts.

## Introduction

Agriculture, wastewater, and fossil fuel use have exceeded global thresholds for nitrogen (N) and phosphorus (P) [1–4], resulting in eutrophication of two-thirds of freshwater ecosystems globally [5–8]. Excess nutrients and other water pollutants such as heavy metals and waste from humans and livestock degrade aquatic ecosystem integrity, leading to trillions of USD in economic damages and the premature death of approximately 2 million people annually [9–12]. Mitigating these global water crises will require improved monitoring and management, which themselves depend on public understanding and financial support [13–16]. Consequently, improving public understanding and interaction with aquatic ecosystems is a planetary priority [17–19].

Because of the high spatiotemporal variability typical of both surface and subsurface aquatic ecosystems [20–22], identifying pollutant sources requires repeated sampling of many locations in the watershed [23–25]. This type of synoptic sampling provides a high-resolution view of water chemistry throughout the network, potentially generating insights into hydrological and biogeochemical processes [26–28], as well as identifying the location and spatial extent of pollutant sources [29–31]. Pollutant sources range in size from discrete point sources, such as a wastewater discharge, to diffuse nonpoint sources, such as runoff from agricultural fields [32, 33]. They also vary in duration, from persistent sources that are always active to intermittent sources that only deliver pollutants to the ground or surface water during certain ecohydrological conditions [34–36].

While measuring or modeling water chemistry continually everywhere in the stream network remains impossible, a suite of ecohydrological metrics have been developed to inform management based on repeated synoptic sampling [1, 23, 27, 29, 37]. For example, the relationship between spatial variability and watershed size can reveal the patch size of processes driving water chemistry, revealing the relative importance of delivery from terrestrial environments and processing in aquatic environments [26, 29, 38, 39]. Likewise, the persistence of spatial patterns in water chemistry through time can reveal changes in pollutant sources or sinks, informing the necessary sampling frequency [23, 27, 40, 41]. In practice, high spatial persistence of a solute in a watershed allows for a single synoptic sampling event to representatively characterize its dynamics [23, 29, 40]. While synoptic sampling requires substantial resources and coordination, combining such metrics with traditional analysis of watershed land use and land cover could improve effectiveness of monitoring and restoration measures. So far, these analyses have primarily been done in temperate and high-latitude ecosystems [23, 27, 38, 40], leaving important unknowns about other ecosystem types.

In semiarid and arid regions, the degradation of aquatic ecosystems has been particularly extreme because of high water demand and low water availability for green, blue, and gray water use [17, 34, 42–44]. This combination often results in intense hydrological and chemical disruption of surface and groundwater in semiarid and arid watersheds, which are often endorheic [17, 42, 43, 45]. In addition to anthropogenic pressure, aquatic ecosystems in semiarid regions are naturally dynamic because of extreme hydrological and biogeochemical variability [46–48]. Changes in precipitation and evapotranspiration result in large expansions and contractions of the surface water network, and wetting and drying cycles create

heterogeneous biogeochemical conditions [49–51]. This combination of human and natural variability could create nutrient source and sink patterns that are distinct from temperate regions, potentially complicating the identification and mitigation of nutrient sources in heavily impacted semiarid watersheds.

In this context, we organized a series of participatory synoptic sampling events in the Utah Lake watershed in the western US. We used a citizen science approach for two reasons. First, waterbodies in this region are experiencing eutrophication and water overallocation [43, 52–54], partly because of a lack of public connection with local aquatic ecosystems [55]. For example, this disconnect has led to a lack of public will to implement wastewater treatment measures that could decrease delivery of bioavailable nutrients to Utah Lake [56, 57] and even the consideration of a radical reengineering of the lake, including large artificial islands [58, 59]. Second, the Utah Lake watershed is nearly 8,000 km$^2$, making traditional synoptic sampling impractical and expensive. By collaborating with local nonscientists, we were able to sample nearly 200 locations throughout the watershed within a few hours, reducing variability from temporal changes in water flow and chemistry [27, 60]. We focused on three major questions: 1. What are the primary sources of carbon and nutrients in the Utah Lake Watershed, 2. How much solute retention or release is there in the surface water network, and 3. What are the general patterns of solute chemistry in these semiarid river networks. Though we did not collect quantitative data on the attitudinal effects of participating in the samplings, we hypothesized that learning about and spending time in the diverse tributaries to Utah Lake would improve public awareness and proclivity to address local environmental issues [16, 18, 61, 62]. Regarding the ecohydrological issues surrounding Utah Lake, we hypothesized that the complex human footprint and variable hydrology in this semiarid watershed would create low spatial persistence of pollutant sources and high critical source area behavior with a few influential areas disproportionately influencing water quality [29, 31, 33]. To test these hypotheses, we analyzed volunteer-collected samples for a broad suite of physicochemical parameters, calculating the ecohydrological metrics we describe below.

## Methods

### The Utah Lake watershed

Utah Lake is one of the largest freshwater lakes in western North America, with a surface area of 375 km$^2$ and a natural watershed area of 7,640 km$^2$ [54]. As a part of the Great Salt Lake watershed, Utah Lake is a remnant of Lake Bonneville, which covered up to 51,000 km$^2$ of what is now Utah, Nevada, and Idaho until about 15,000 years ago [63–65]. The watershed ranges from 1,368 to 3,586 MASL and is characterized by relatively pristine high-elevation headwaters in the Wasatch Mountains—although mining, livestock grazing, rural subdivisions, and ski resorts are present. Low elevation valleys have a mix of urban and suburban development interspersed with irrigated agricultural land, with an overall human footprint of up to 93% in the valley bottom. Utah has one of the fastest growing populations in the US [66], and agricultural land is increasingly being converted for suburban development. Seven wastewater treatment plants, serving approximately 600,000 people in the valley region, discharge treated effluent into tributaries to the lake (S1 Fig).

In the watershed headwaters, the dominant geology is limestone and quartz [67]. The vegetation consists of mixed aspen, conifer, and maple forests at high elevations, transitioning to scrub oak and sagebrush at lower elevations. The high-elevation hydrology is complex, including gaining reaches (i.e., net flow of subsurface water into surface water flows) and losing reaches (i.e., net flow into subsurface water), particularly in areas of karst conduits and colluvial materials [68]. At the base of the mountains, where rivers flow into Lake Bonneville

sediment deposits at the outer perimeter of the prehistoric lake, reaches become primarily losing, and water transport occurs largely through shallow groundwater flowpaths [69]. Near the lake, streams once again become gaining, mixing with new streams generated by natural and artificial recharge that feeds many springs flowing into the lake [70, 71].

## Sampling design

We classified the subwatersheds into one of four categories based on land use and hydrologic condition (Table 1): Agricultural unregulated (Spanish Fork River), Mixed dammed (Provo River), Mountain urban (American Fork River and Hobble Creek), and Valley tributaries (Benjamin Slough, Goshen Valley, Mill Race, and others). The Agricultural unregulated subwatersheds have been slower to experience rapid population growth and still remain mostly under agricultural uses. One subwatershed (Diamond Fork) receives artificially diverted flow from the nearby Strawberry River, but for most of the Agricultural unregulated subwatersheds, the in-stream hydrologic modifications are minimal. All the categories have some degree of flow infrastructure (e.g., check dams, channelized reaches, dikes, etc.), but Mixed dammed subwatersheds include two large reservoirs (capacities of 0.18 and 0.39 km$^3$, respectively) that drastically alter the downstream hydrology. Land use in Mixed dammed subwatersheds is a mixture of both agriculture and urban. Mountain urban subwatersheds include high contrast land use and degree of impact, with mostly pristine headwaters before entering highly developed land use and consequently modified

**Table 1. Watershed characteristics for contributing areas to Utah Lake watersheds.**

| Category / River name | Area (km$^2$) | Mean Elevation (MASL) | Mean specific discharge (L sec$^{-1}$ km$^{-2}$)[a] | Forest (%)[b] | Herbaceous (%)[c] | Developed (%)[b] | Impervious (%)[b] | # of Sites |
|---|---|---|---|---|---|---|---|---|
| **Agricultural unregulated** | | | | | | | | |
| Spanish Fork | 1725 | 2137 | 2.26 | 55.9 | 5.5 | 1.9 | 0.5 | 43 |
| **Mixed dammed** | | | | | | | | |
| Provo | 1774 | 2320 | 3.32 | 64.9 | 3.5 | 5.5 | 1.1 | 112 |
| **Mountain urban** | | | | | | | | |
| American Fork | 160 | 2493 | 1.44 | 69.7 | 3.5 | 3.9 | 1.3 | 25 |
| Hobble Creek | 298 | 2158 | 2.59 | 58.4 | 7.5 | 2.1 | 0.6 | 22 |
| **Valley tributaries** | | | | | | | | |
| Mill Race | 47 | 1899 | 27.8 | 47.1 | 2.9 | 41 | 23 | 6 |
| Dry Creek | 111 | 2048 | 0.32 | 40.5 | 7.7 | 20 | 7.0 | 4 |
| Goshen Valley | 1355 | 1844 | - | 33.5 | 7.4 | 4.5 | 0.8 | 7 |
| Benjamin Slough | 326 | 1771 | 4.40 | 37.6 | 7.0 | 10 | 3.6 | 23 |
| Other | 906 | - | - | - | - | - | - | 12 |
| **West desert** | | | | | | | | - |
| Cedar Valley | 665 | 1664 | - | 18.4 | 13 | 1.0 | 0.4 | |
| Tickville Gulch | 157 | 1957 | - | 44.8 | 9.2 | 3.6 | 1.4 | |
| Lake Mountains | 116 | 1631 | - | 18.5 | 8.1 | 5.8 | 2.2 | |
| **Utah Lake (total)** | 7640 | 1990 | 2.16 | 43.8 | 6.2 | 10 | 2.1 | |

Subwatersheds were delineated using the application USGS StreamStats. Where categories represent multiple subwatersheds, statistics for the major contributors are given.

[a]Average annual discharge from 1980–2003 (PSOMAS 2007).

[b]2011 National Land Cover Database (NLCD).

[c]1992 NLCD.

stream channels at low elevations. Valley tributaries include the groundwater sourced tributaries that originate near the lake as described above. These subwatersheds also include return flow from agricultural water diversions, and effluent from wastewater treatment plants and other industrial facilities. The Utah Lake watershed also includes the West desert region (Cedar Valley, Tickville Gulch, and Lake Mountain), which has only small, remote ephemeral springs that are largely inaccessible. These subwatersheds were excluded from sampling because they are not connected to the lake via surface water.

We initially selected 500 sampling sites from the Ambient Water Quality Monitoring System (AWQMS), a database curated by the Utah Division of Water Quality, and the Utah State University Water Quality Extension citizen science program, Utah Water Watch. We used a clustering technique by including sites just above and below a confluence to maximize watershed coverage and minimize travel distances. We consolidated the initial 500 sites to 270 by merging redundant locations and removing inaccessible sampling locations (e.g., difficult terrain or private ownership) and sites with no surface flow even during snowmelt. Nearly all remaining sites were publicly accessible, and for the few locations on private property, we obtained verbal permission from landowners. We used the application USGS StreamStats to delineate watersheds, calculate watershed areas ($km^2$), and estimate land use and land cover from the National Land Cover Database (NLCD) for 1992 and 2011. Land use was classified as forested, developed, impervious surface, or herbaceous upland for each watershed.

## Participatory science

The practice of involving nonprofessionals in research (i.e., participatory or citizen science) has been extensively used in the natural and social sciences [16, 72, 73]. While there are trade-offs to participatory approaches, including less control over data acquisition and costs for participants, there can be substantial community and scientific benefits [73–75]. On the community side, citizen science can create educational opportunities and foster trust between researchers, regulators, policymakers, and the public [74, 76–78]. On the scientific side, citizen science can provide opportunities and value for researchers, including enabling novel experimental designs (e.g., the collection of hundreds of samples synchronously) and informing research priorities by improving researchers' awareness of local needs and policy priorities [10, 75, 76, 79].

We collaborated with an undergraduate course on watershed ecology to develop a multi-year participatory science program. This program included 2 public lectures, 7 community events (e.g., fairs, festivals, and educational evenings) where we made presentations or staffed interactive booths, and 6 synoptic samplings. From the beginning of the project, we partnered with existing water organizations, including the Provo River Watershed Council, Utah Water Watch, the Utah Lake Commission, the Utah Division of Water Quality, and the Utah Valley Earth Forum. These partnerships were pivotal in recruiting volunteers, designing the samplings, and disseminating the results. We additionally advertised through social media (Facebook, Twitter, and Instagram), email lists (including past participants), online videos, and paper fliers for approximately one month before each of the six sampling events. These advertisements targeted local university students, youth groups, environmental groups, and the broader community. At the community events, we presented a model watershed (EnviroScape, Herndon, VA) to demonstrate runoff and transport of pollutants represented by food coloring. In all our interactions with the public, we emphasized the historical context of human-water interactions in the watershed, the current ecological status of the lake system, and potential policy and personal actions to improve the health of the lake.

## Sampling events

The synoptic sampling events were the central experiences for volunteers in the program. We carried out six sampling events in 2018 and 2019, but due to COVID-19 and other delays, the chemical analyses of the last three samplings have not yet been finalized. We report the results of the 2018 March (Spring), July (Summer), and October (Fall) samplings in this paper. Volunteers were invited to sign up in advance for the sites they wanted to sample on the project website (https://utahlakecollab.wixsite.com/utahlakecollab), choosing from an interactive online map of the 270 locations. This planning process encouraged volunteers to explore the entire watershed virtually before the event, looking at areas where they already had experiences and imagining locations they had not yet visited.

To reduce variability from sampling error, we used careful training, simplification of procedures, and replication of sampling (i.e., we asked multiple volunteers to collect samples from the same site so we could quantify sampling error). Other citizen science studies have found that when such precautions are taken, the benefit of using volunteers to collect large numbers of samples outweighs the tradeoffs of this kind of approach [80–82]. On the days of the samplings, we distributed informational flyers while volunteers waited to collect sampling kits and directions to sites. The flyers described the sampling procedure and provided context about water use and water pollution in the area (S2 Fig). Before distributing the sampling kits, we trained each volunteer individually, showing them how to collect water samples safely and reproducibly. We provided simple field sheets, where participants recorded site conditions and bottle numbers. After completing the sampling, participants returned their samples and datasheets to the distribution locations, where we performed quality control, noting samples with incomplete data or other irregularities (e.g., broken filters, partially filled bottles).

Because this research involved nonprofessional volunteers, we consulted the Institutional Review Board (IRB) at Brigham Young University (BYU), which oversees all research involving human subjects. We were informed that IRB approval was not needed because volunteers were not the subject of the research (i.e., we did not collect information about their identities or experiences). This limited our ability to quantitatively assess the community outcomes of the research but made participation less burdensome and improved inclusivity, particularly for members of the community unwilling to share personal information for philosophical or political reasons.

## Laboratory analyses

Samples were filtered in the field with pre-rinsed 0.45 μm cellulose acetate filters into 60 ml high-density polyethylene bottles and immediately frozen or refrigerated until analysis (typically within 2 weeks of sampling). Anions ($NO_3^-$, $NO_2^-$, $SO_4^{2-}$, $Cl^-$, and $PO_4^{3-}$) and cations ($NH_4^+$) were quantified by ion chromatography (Dionex Thermofisher HPIC). Soluble reactive phosphorus was quantified colorimetrically using the ascorbic acid method [83], reported hereafter as $PO_4^{3-}$ for simplicity. Dissolved organic carbon (DOC) and total dissolved nitrogen (TDN) were quantified using a C/N auto-analyzer (Elementar, Langenselbold, Germany). Dissolved inorganic nitrogen (DIN) was calculated as the sum of $NO_3^-$, $NO_2^-$, and $NH_4^+$ from the ion chromatography.

## Ecohydrological metrics and statistical analyses

We calculated three recently developed ecohydrological metrics that indicate solute dynamics: spatial variance thresholds, subwatershed leverage, and spatial persistence [27, 29]. Spatial variance thresholds indicate the predominant spatial scale where solute delivery or removal is occurring, analogous to the representative elemental area concept [84, 85]. For each of the

main tributaries (Table 1), we plotted the scaled (subtracted the mean and divided by the standard deviation) concentration for each solute against subwatershed area. Changes in variance with spatial scale can be caused by mixing of tributaries, in-stream processing, and changes in terrestrial-aquatic linkages. We used the pruned exact linear time (PELT) method to test for changes in variance, implemented in the changepoint package of R [86]. We hypothesized that the variance would decrease with spatial scale, following observations from other river networks [29, 38].

We calculated subwatershed leverage by multiplying the difference in subwatershed and watershed outlet nutrient concentrations with the ratio of subwatershed area to watershed outlet area [27, 29]. Assuming similar specific discharge throughout the watershed, leverage indicates the amount of flux at the outlet that can be explained by the contribution of each subwatershed [29, 38]. Positive leverage indicates the watershed is a net sources of the given solute and negative leverage indicates the watershed is a net sink. Very large leverage values can occur—for example, well over 100%—because some material is retained or removed as the water flows over and through the landscape before reaching the outlet. Thus, sites with large leverage values (i.e., >100%) can be considered highly influential contributors to solute concentrations at the outlet. Recently, mean leverage values across subwatersheds have been used to infer network-scale production or retention of solutes [27]. When the watershed leverage mean is positive, this implies there has been nutrient removal within the surface water network (i.e., there is more solute in the tributaries than can be accounted for at the outlet), whereas a mean negative leverage value indicates production within the surface water network. Valley tributaries subwatersheds were excluded from leverage analyses because they did not converge within a network to a single outlet (i.e., many discharged directly into Utah Lake).

To assess the consistency of observed spatial patterns, we calculated spatial persistence with Spearman's rank correlation [23, 29]. This analysis calculates a correlation coefficient ($\rho$) of solute concentrations ranked from highest to lowest for each category and solute, across each pair of Seasons (i.e., Spring-Summer, Summer-Fall, and Spring-Fall). We graphed means and ranges of $\rho$ values and tested for the effects of Season and land use category on solute persistence using ANOVA.

We also used parametric statistics to compare links between land use and land cover with water chemistry for the main tributaries. For each solute, we tested for differences among watersheds with Analysis of Variance (ANOVA). We tested for links between land use and catchments characteristics with water chemistry using generalized least squares models. The models used concentration of each select solute ($PO_4^{3-}$, DIN, TDN, DOC, $Cl^-$, and $SO_4^{2-}$), including season and land use (e.g., developed, impervious, forest, and herbaceous) as independent variables. A second order Akaike Information Criterion (AICc) was calculated for each possible subset model, then the subset model with the smallest AICc score was selected as the final model (reported below).

We performed all statistical analyses in R, using the dplyr and ggplot2 packages [87–89]. We used ArcGIS Pro (ESRI) to map solute concentrations, using open-source base layers. Sample concentrations, including geographic information, and code used in analysis are available at https://doi.org/10.4211/hs.85f3584dccb54afba5698ac615ff949a.

## Results

In 2018 and 2019, our outreach efforts reached approximately 6,500 community members directly. Across the first three sampling events (the ones reported in this paper), we had over 150 unique participants, with at least a third of participants attending more than one event. Most participants sampled in small groups although some worked alone. Detailed

demographic data was not collected for participants in this phase of the project, however most attendees were estimated to be between 18 to 30 years old, with ages ranging from less than 1 over 80 years old. When informally asked about their experience, responses were overwhelmingly positive, including statements like, "I had no idea there were so many beautiful streams in Utah Valley," and, "this opened my eyes to how much we depend on this water. It actually comes from somewhere before my sink!"

Most sampling kits were used correctly (e.g., bottles filled with filtered sample), with <10% of the samples rejected because of user error. Incomplete or illegible labeling was the most common error, though there were a few instances of complete communication breakdown (e.g., one bottle was returned filled with soil).

## Solute concentrations

The spatial distribution of solute concentrations across the watershed is shown in Fig 1. This map includes data for samples collected from Utah Lake, which were excluded from other analyses. Point color corresponds with Utah's numeric water quality standards for N and P (N > 4 mg L$^{-1}$ and P > 0.05 mg L$^{-1}$) and the 25$^{th}$ and 75$^{th}$ percentile of measured concentrations for other parameters. DOC concentration varied relatively evenly across the watershed, while DIN, TDN, $PO_4^{3-}$, $SO_4^{2-}$, and $Cl^-$ concentrations were highest at sites near or on Utah Lake (Tukey, $p$-adj.<0.001).

Solute concentrations were different across the three sampling seasons (Spring, Summer, and Fall) through the year (ANOVA, $F$-stat = 5.896, $p$-value<0.01) and across categories (ANOVA, $F$-stat = 96.055, $p$-value<0.0001) for all solutes (Fig 2 and S1 Table). Pairwise analysis determined that solute concentrations in Spring were higher than Summer or Fall (Tukey, $p$-adj. = 0.002), but also depended on land use category (ANOVA, $F$-stat = 4.34, $p$-value<0.001). A pairwise analysis across all categories determined that DOC concentration was higher in Summer than Fall or Spring (Tukey, $p$-adj<0.001), and TDN concentration was higher in Fall than during Summer or Spring (Tukey, $p$-adj<0.05). All solute concentrations were higher at Valley tributaries sites than the other three categories ($p$-adj.<0.001), and Mountain urban were higher than Mixed dammed ($p$-adj = 0.005).

Regressions of solute concentration by Season and land use found that % impervious surface was positively correlated with higher concentrations for all solutes except DOC (Table 2). In regression models of TDN, DOC, and $SO_4^{2-}$, backward exclusion criteria found that only Season was significant. In regression models of $SO_4^{2-}$ and $Cl^-$, backward exclusion criteria found that only % herbaceous upland was significant. Correlation coefficients ($R^2$) for the models were between 0.11 and 0.25.

## Leverage and spatial persistence

In general, scaled concentrations and leverage by watershed area for the different solutes did not show a funnel shape that is typical of humid and temperate watersheds (Figs 3 and 4A) [38]. Instead, many of the solutes exhibited hourglass shapes, with higher scaled concentration and leverage at the largest subwatershed size. However, the funnel shape did occur for DIN concentrations in the valley tributaries, which had particularly high leverage in the smaller catchments (<100 km$^2$). Variance thresholds (km$^2$), calculated from the scaled concentration data using the PELT method, are listed in S3 Table. Only two solutes (DOC and TDN) were calculated to have a single changepoint: TDN, 0.78 km$^2$, and DOC, 0.62 km$^2$ (although DOC concentrations showed no variance collapse). PELT calculated two variance thresholds for both $Cl^-$ (3.69, 666.00) and $SO_4^{2-}$ (62.6, 666.0), each having one changepoint closer to the headwaters and one nearer to the lake. DIN had three variance thresholds, all of which were

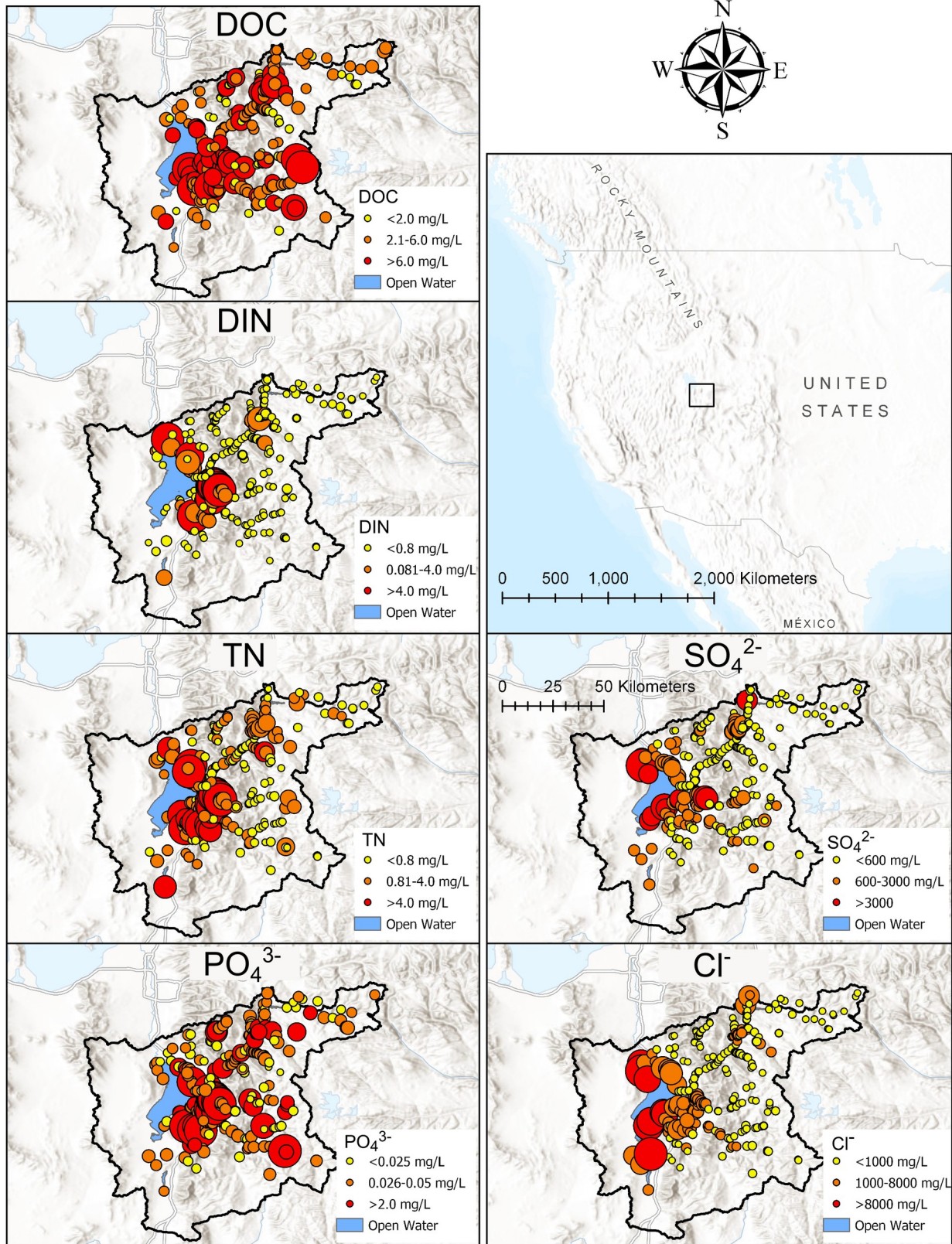

**Fig 1. Maps showing the concentration of solutes across the Utah Lake watershed.** Samples were collected from the watershed (outlined in black) and averaged across three citizen science synoptic sampling events. Point color represents numeric water quality standards for N and P

values, or 25th and 75th percentile for others. Point size is scaled to concentration. Basemap source: USGS National Map and Earth Resources Observation and Science Center.

relatively small subwatershed sizes (0.78, 14.3, 18.0). Six changepoints were calculated for $PO_4^{3-}$ (0.23, 10.3, 10.9, 14.3, 18.0, 644), with five out of the six thresholds being found at small subwatershed sizes. Occasional outliers (single points at mid-range and large watershed size) may have had an oversized effect on the overall pattern.

Most subwatersheds exhibited moderate to low leverage (i.e., < ±25% leverage) on watershed outflow concentrations of $Cl^-$, and $SO_4^{2-}$ (85, and 84% of watersheds respectively; Figs 4B and 5 and S2 Table). DOC, N, and P dynamics were much more concentrated (i.e., showing a critical source area behavior), with around two-thirds (59–69%) of watersheds having a moderate to low effect on outflow concentrations and the remaining third controlling flux. There were many more highly influential subwatersheds (i.e., >100% leverage) for TDN and $PO_4^{3-}$ than for DOC, $Cl^-$, and $SO_4^{2-}$, in line with the expected pattern from discrete sources of N and P in the watershed. Unexpectedly, there were very few highly influential subwatersheds for DIN concentrations, potentially due to the very high relative flux of DIN at the watershed outlet (i.e., large loads in subwatersheds still remain <100%). The number of subwatersheds that

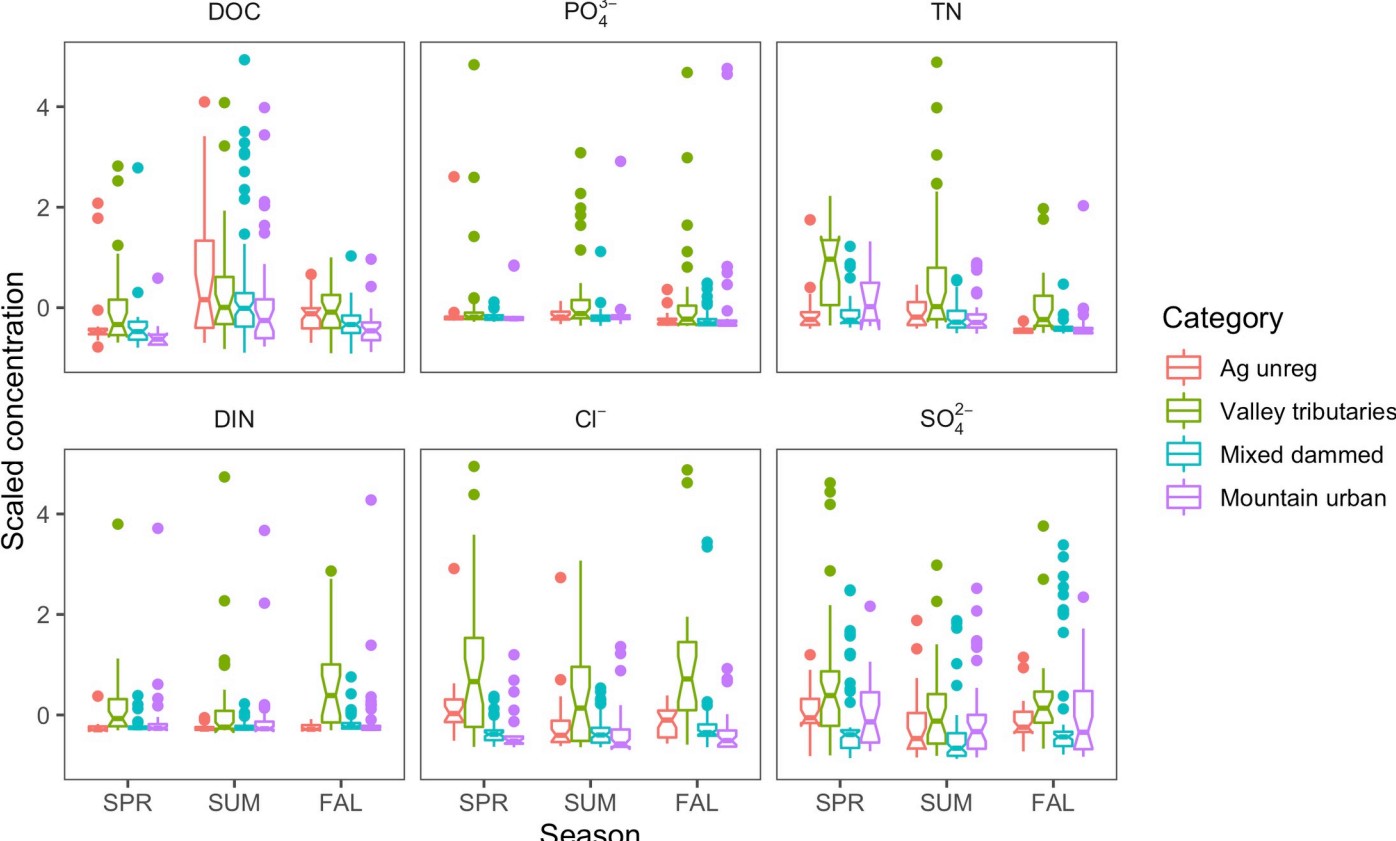

**Fig 2. Scaled concentration of solutes in Utah Lake subwatersheds.** Samples were collected during synoptic sampling events in three seasons (SPR = Spring, SUM = Summer, FAL = Fall) in watersheds with different land use categories (Agricultural unregulated, Mixed dammed, Mountain urban, and Valley tributaries). Boxplots represent the 25th, 50th, and 75th percentiles, points within 1.5 times the interquartile range, and points beyond. The notches represent the 95% confidence interval of the median (non-overlapping notches suggest statistically significant differences between populations).

**Table 2. Multiple linear regression models of solute concentration by land use and Season.**

| Solute | Model variables | $R^2$ |
|---|---|---|
| DOC | Season*** + % forest*** + % impervious*** | 0.192 |
| $PO_4^{3-}$ | % impervious*** | 0.147 |
| TDN | Season*** + % impervious*** | 0.231 |
| DIN | % impervious*** | 0.245 |
| $Cl^-$ | % herbaceous* + % impervious*** | 0.123 |
| $SO_4^{2-}$ | Season + % herbaceous*** + % impervious*** | 0.115 |

Solute concentration (DOC, $PO_4^{3-}$, TDN, DIN, $Cl^-$, and $SO_4^{2-}$) was regressed by land use (% forest, % developed, % impervious surface, % herbaceous upland) and Season (Spring, Summer, and Fall) as independent variables. Final models were selected as those with the smallest AICc score for the respective solute. Sign indicates correlation (+ = positive,— = negative). Asterisks denote significant $p$-values:

'***' < 0.001;

'**' <0.01;

'*' < 0.05; ' ' < 0.1.

were highly influential (i.e., >100% leverage) ranged from 11–16% for TDN and $PO_4^{3-}$, 8% for DOC, and only 1–2% for DIN, $Cl^-$, and $SO_4^{2-}$.

Mean leverage values for each solute and subwatershed category further illustrated the dynamic behavior of N and P in the spring and summer in comparison with DOC, $Cl^-$, and $SO_4^{2-}$ (Fig 5). For example, TDN exhibited a strong removal signal (mean leverage >0) for all categories in the spring and summer, with means closer to 0 indicating a conservative transport (0 net production or removal) in the fall. DIN exhibited a weak production signal in all categories and seasons. $PO_4^{3-}$ varied considerably by category and season. In contrast, $Cl^-$ and $SO_4^{2-}$ maintained consistent neutral mass balances across category and season. DOC switched from a net source to a sink from spring to summer in the Unregulated agricultural subwatersheds, but remained neutral for the Mountain urban subwatersheds, and maintained a mostly consistent sink capacity in the Mixed dammed subwatersheds.

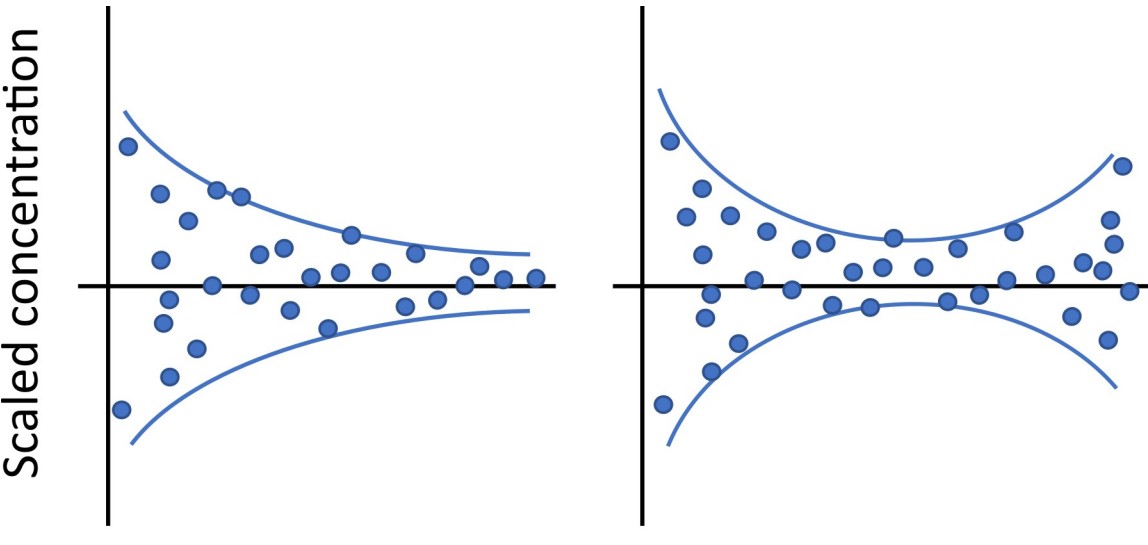

**Fig 3.** Theoretical diagram of spatial variability collapse (left) and spatial variability pattern observed in Utah Lake watershed (right).

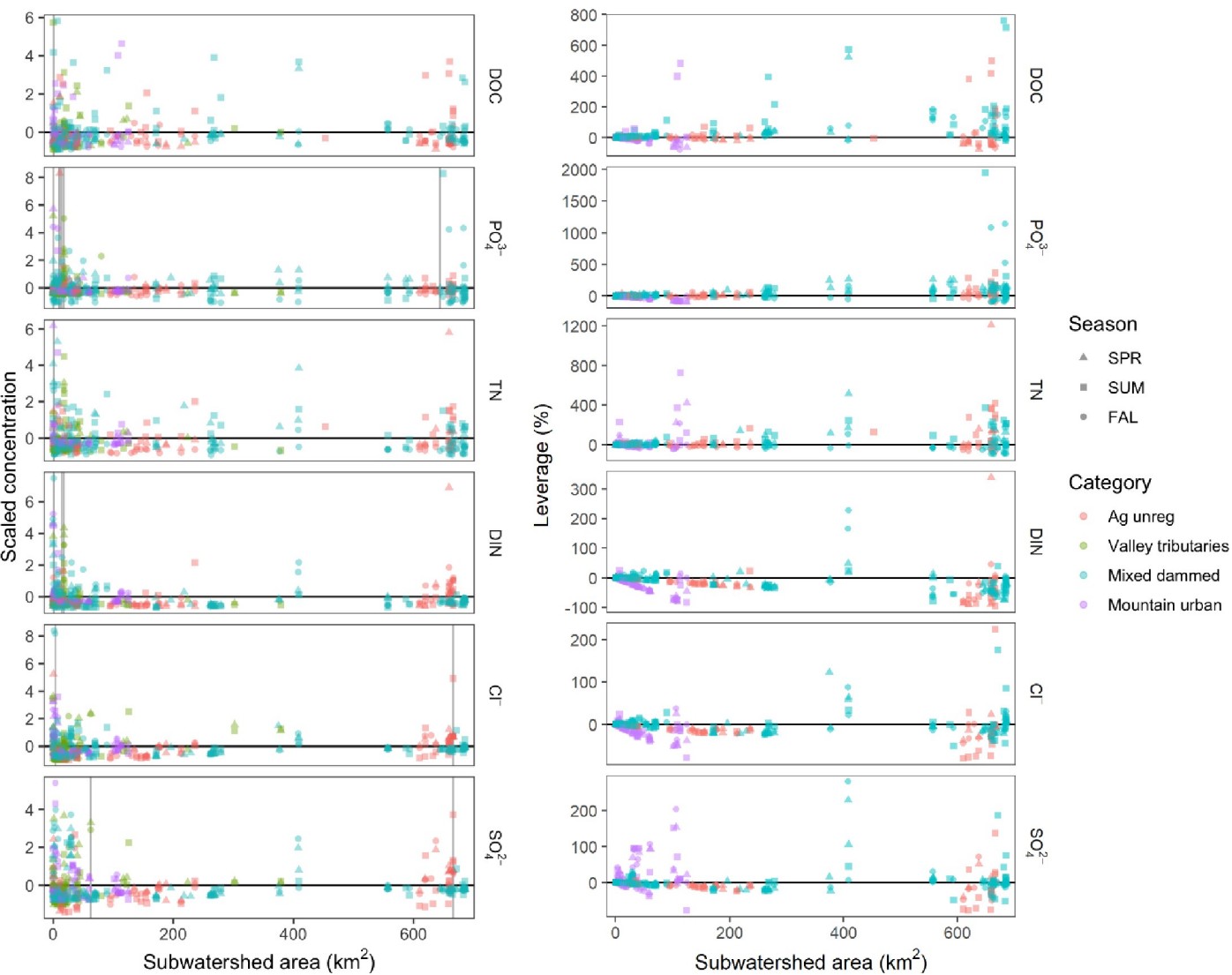

**Fig 4.** Scaled solute concentration (a, left) and leverage (b, right) by watershed area for solutes of interest (DOC, PO₄³⁻, TDN, DIN, Cl⁻, and SO₄²⁻) for sites within four land use and hydrologic categories of the Utah Lake watershed. Samples were collected on synoptic sampling events conducted in three seasons: Spring, Summer, and Fall of 2018. Horizontal lines represent the means of the raw concentration data for that particular solute; vertical lines represent change points detected by PELT analysis.

Spatial persistence was specific to solute and land use (Fig 6 and S3 Table). DOC and $PO_4^{3-}$ had lower spatial persistence (0–0.5) than the other solutes (0.3–1). $Cl^-$ and $SO_4^{2-}$ had the highest spatial persistence (>0.7). DIN and TDN had intermediate levels of persistence (0.5–0.7). Persistence was highest overall at Mountain urban and lowest in Valley tributaries and Agricultural unregulated subwatersheds, although the order of persistence as dependent on solute (ANOVA, $F$-stat = 3.514, $df$ = 30, $p$-value<0.001).

## Discussion

### Novel spatial hydrochemical patterns

Our study emphasizes the unique hydrochemistry of semiarid and mixed natural-urban regions. In our results, $PO_4^{3-}$ was less spatially stable than other major ions, which was also the

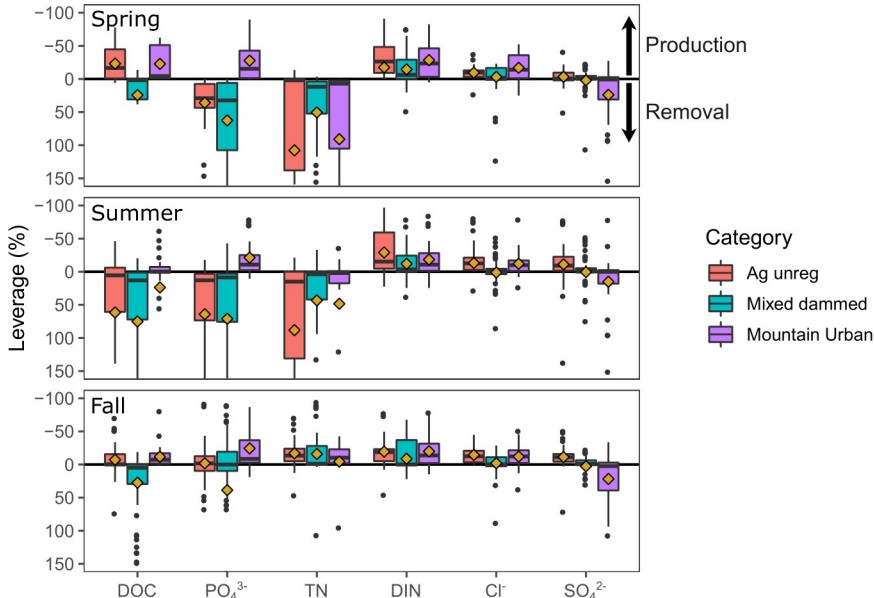

**Fig 5. Distribution of leverage values show prevailing watershed production or removal for solutes across seasons.** Diamonds represent the mean leverage value for each subwatershed category. The horizontal black line at y = 0 represents a neutral mass balance. Diamonds that are above the black line indicate solute production and diamonds below the black line indicate solute removal within the surface water network.

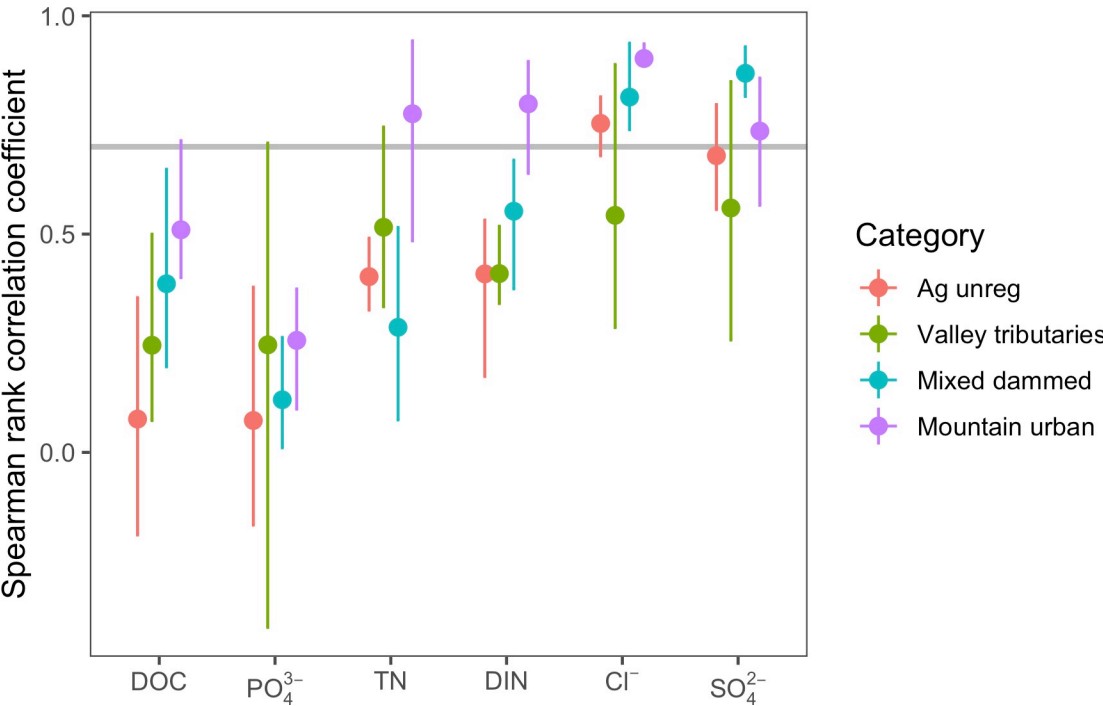

**Fig 6. Spatial persistence for solutes (DOC, PO43-, TDN, DIN, Cl⁻, and SO42-) for different categories of land use within the Utah Lake watershed.** Spatial persistence is calculated as a pairwise Spearman rank correlation coefficient ($\rho$) among synoptic samplings from three seasons (Spring, Summer, and Fall). Points represent mean $\rho$ for all pairwise comparisons within each category, and bars give the range of values. The horizontal grey line marks $\rho = 0.7$; indicating that the majority of the spatial pattern was retained between the two samplings.

case in temperate agricultural watersheds [23, 29] and in Arctic tundra watersheds [27]. The low spatial persistence of $PO_4^{3-}$ compared with other solutes like $Cl^-$ suggests two non-exclusive factors: one, sources other than natural geologic weathering introduce variability and two, biological processes involving P uptake are variable. For example, changes in hydrological connectivity across seasons and storm events can alter the delivery and chemical availability of $PO_4^{3-}$ and other nutrients [90–93]. Likewise, P is often a limiting nutrient for some subwatersheds, introducing another source of variability as biological demand may exceed supply [36, 94, 95].

Valley tributaries stood out as having particularly low spatial persistence across all solutes (Fig 6). We hypothesize that this is at least partially a function of the natural difference in solute concentrations between dilute snowmelt stream water and solute-rich groundwater, combined with changes in source based on hydrologic conditions [69]. When discharge is high, stream water diluted with snowmelt flows farther into the valleys before the change from losing to gaining occurs. During low flow, the switching point of losing to gaining moves upstream, creating large variability in solute concentrations observed at valley sites between the mouth of the canyons and the lake.

Our analysis did not return a single spatial threshold for variance collapse in solute concentration, unlike in other studies [50, 96–98]. For example, Northern Boreal watersheds tend to have variance collapse in DOC at 15 $km^2$ [40]. In Arctic tundra, the threshold for DOC and $NO_3^-$ is similar (10–20 $km^2$), and $PO_4^{3-}$ is slightly larger (25 $km^2$) [27]. Mined Kentucky headwaters have variance collapse in major anion and cation concentrations between 15 and 75 $km^2$ [99]. DOC, $PO_4^{3-}$, TDN, and DIN all had threshold values at very small subwatershed areas ($<1$ $km^2$). These low thresholds are likely influenced by point sources, such as wastewater effluent and field drains, which we reported as subwatershed areas of 0 $km^2$. $PO_4^{3-}$, $Cl^-$, and $SO_4^{2-}$ had thresholds at small and large subwatershed sizes. The hourglass pattern in spatial variance may be also be impacted by the fact that the average subwatershed areas for the Valley tributaries and Mountain urban were much smaller (40 and 32 $km^2$, respectively) than the Mixed dammed and Agriculture unregulated (200 and 281 $km^2$, respectively). The lack of variance collapse in our study, like the low spatial persistence, may have been due to increased solute concentrations at groundwater-influenced sites near the lake [69]. Alternatively or additionally, we may have detected differences in spatial variance at finer scales because of the high-resolution sampling (i.e., such variability could exist in other areas but not have been detected because of coarser spatial sampling).

## Urban sources affect all solutes except DOC

This study identified hot spots of solutes of concern (Fig 1), and determined that position within watershed is important in determining concentration dynamics [100]. $PO_4^{3-}$, TDN, DIN, $Cl^-$, and $SO_4^{2-}$ concentrations were highest in Valley tributaries and other low-elevation reaches, and solute removal or dilution occurred at mid-elevation reaches, indicated by the map of concentrations (Fig 1) and narrowing of solute concentrations in mid-range subwatershed sizes (Fig 4A). Impervious surfaces contributed to overall higher solute concentrations (Table 2), but solute concentrations were more variable in reaches with agricultural activity (Fig 6). This could be due to both the direct effect of impermeability on nutrient export (e.g., stormwater drainage) as well as a correlation between percent impermeability and the presence of wastewater effluent [34, 101]. We note that our linear models had low predictive capability (up to 25%), suggesting the need for additional explanatory metrics (e.g., geology and updated land cover data). Decreases in solute concentrations at sites in the valley could be due to losses to groundwater [69] or sorption of P to Lake Bonneville sediments [54].

Valley tributaries were significant sources of N, a majority of which was DIN (Figs 2 and 4A). Multiple linear regression analyses showed that models of TDN and DIN had higher correlation with % impervious surface than any of the other solutes (Table 2), and both categories of N had lower persistence across Valley tributaries than the other subwatershed categories. Valley tributaries, which had the highest percentage of developed land use (Table 1), had the highest variability in N concentrations. Biological activity could also be responsible for high variability and seasonal differences in N concentrations, similar to what was observed in other mountainous western US watersheds [102].

This study confirms that urban point sources may disrupt spatial variability collapse. For example, in highly urbanized watersheds in New York, $NO_3^-$ decreased in variability with increased watershed size, but not SRP or $NH_4^+$ [94]. We found DIN increased below wastewater treatment plants in Mixed dammed and Mountain urban watersheds, although in Mixed dammed subwatersheds, DIN is subsequently diluted and/or removed at downstream sites (Fig 1). $PO_4^{3-}$ concentration exceeded State of Utah numeric criteria for nutrient pollution throughout all subwatershed categories and sizes, suggesting that point and non-point sources (e.g., stormwater, agricultural water, and natural geological deposits) contribute to elevated P in local streams.

## Citizen science

In this study, we provided the opportunity for thousands of local citizens to learn more about point and non-point sources of water pollution in a deep and meaningful way [18]. This engagement has the possibility of creating public support for efforts to address water quality in the Utah Lake watershed [103]. Future directions of this work include using educational research tools to quantify the impact of participation on knowledge, attitude, and behavior. This study demonstrates that citizen scientists can help professional researchers accomplish study methodologies that are otherwise prohibitive. This has the dual benefits of extending capacity for scientific observation, and fundamentally changing public awareness and mentality. Both benefits subsequently influence how water resources are managed [61]. In this sense, participatory water quality monitoring is not only a means of increasing understanding of how water and nutrients propagate through watersheds; it is a mechanism to improve water quality itself and encourage sustainable stewardship [79, 80].

## Conclusion

Our results demonstrate the high spatial and temporal variability of $PO_4^{3-}$ within this watershed. However, at intermediate subwatershed sizes, $PO_4^{3-}$ removal or dilution occurred (Fig 4A). Point sources and groundwater around the lake contributed N in the form of DIN, and, like $PO_4^{3-}$, decreased in concentration in mid-range subwatershed size. In addition to high solute concentrations, Valley tributaries had low spatial persistence, indicating temporally dynamic changes in sources and sinks of solutes. Even though there are inputs from natural (e. g, geology) and anthropogenic (e.g., mining and grazing) sources that contribute to variability in solute concentrations in the headwaters, the variability decreases at intermediate watershed sizes as these diverse headwaters mix. It is at the highly impacted, urban reaches (Valley tributaries and subwatershed sites >500 km$^2$ that normal solute behavior is affected to the degree that variability unexpectedly increases.

The Utah Lake watershed is fundamentally different in network-scale hydrochemistry than previously described temperate, urban, and Arctic watersheds, due to the unique hydrology and human impacts, especially in the lower reaches of the watershed. We encourage including more semiarid regions, specifically endorheic basins, in hydrologic studies because

understanding their distinctive hydrologic characteristics is critical to preserving these unique ecosystems, many of which are threatened by climate change and human development [43].

## Supporting information

**S1 Fig. Map of Utah Lake watershed sites synoptically sampled.** Points colored by land use and hydrologic modification category (red = Agricultural unregulated, green = Valley tributaries, blue = Mixed dammed, purple = Mountain urban). Yellow triangles represent wastewater treatment plants. Basemap source: USGS National Map and OpenStreetMap.
(TIF)

**S2 Fig. Documents provided to citizen science synoptic sampling participants.** Includes sampling instructions and datasheet for recording sample information. Instruction sheet shows photo of co-author GML demonstrating sampling technique.
(DOCX)

**S1 Table. ANOVA test comparing solute concentrations in streams from different land use categories over three synoptic sampling events in the Utah Lake watershed.**
(DOCX)

**S2 Table. Distribution of leverage values in each subwatershed for each solute.**
(DOCX)

**S3 Table. ANOVA test comparing spatial persistence values for solutes measured in different land use categories over three synoptic sampling events in the Utah Lake watershed.**
Asterisks denote significant $p$-values: '***' < 0.001; '**' <0.01; '*' < 0.05; '.' < 0.1.
(DOCX)

## Acknowledgments

We would like to thank Greg Carling, Paul Frandsen, Courtney Fulton, Blair Hansen, Jordynn Scheuller, Hannah Scow, Lindsey Steinhorst, Jesse Morris, and Julianne Capito for contributing to the initial project design; Brian Brown and Isaac St Clair for assisting with data compilation; and the students of PWS 306 in 2019 who created the watershed delineations in USGS StreamStats. We recognize and thank all the participants and volunteers throughout 2018 and 2019, without whom this project would not have been possible.

## Author Contributions

**Conceptualization:** Erin Fleming Jones, Rebecca J. Frei, Jordan D. Maxwell, Rhetta Shoemaker, Andrew P. Follett, Gabriella M. Lawson, Madeleine Malmfeldt, Rachel Watts, Zachary T. Aanderud, Carter Allred, Madeline Buhman, Hunter Burbidge, Amber Call, Isabella Errigo, Natasha A. Griffin, Neil C. Hansen, Jansen C. Howe, Emily L. Meadows, Elizabeth Kujanpaa, Monterey L. Nelson, Elysse Ostlund, Nicholas J. Suiter, Kaylee Tanner, Joseph Tolworthy, Maria Camila Vargas, Benjamin W. Abbott.

**Data curation:** Erin Fleming Jones, Rebecca J. Frei, Raymond M. Lee, Rachel Watts, Amber Call, Trevor Crandall, Leslie Lange, Kaylee Tanner, Benjamin W. Abbott.

**Formal analysis:** Erin Fleming Jones, Rebecca J. Frei, Raymond M. Lee, Jordan D. Maxwell, Trevor Crandall, Benjamin W. Abbott.

**Funding acquisition:** Erin Fleming Jones, Zachary T. Aanderud, Neil C. Hansen, Benjamin W. Abbott.

**Investigation:** Erin Fleming Jones, Rebecca J. Frei, Jordan D. Maxwell, Rhetta Shoemaker, Andrew P. Follett, Gabriella M. Lawson, Madeleine Malmfeldt, Rachel Watts, Carter Allred, Allison Tuttle Asay, Madeline Buhman, Hunter Burbidge, Amber Call, Trevor Crandall, Isabella Errigo, Natasha A. Griffin, Jansen C. Howe, Emily L. Meadows, Elizabeth Kujanpaa, Leslie Lange, Monterey L. Nelson, Adam J. Norris, Elysse Ostlund, Nicholas J. Suiter, Kaylee Tanner, Joseph Tolworthy, Maria Camila Vargas, Benjamin W. Abbott.

**Methodology:** Erin Fleming Jones, Rebecca J. Frei, Jordan D. Maxwell, Rhetta Shoemaker, Andrew P. Follett, Gabriella M. Lawson, Madeleine Malmfeldt, Rachel Watts, Carter Allred, Madeline Buhman, Hunter Burbidge, Amber Call, Isabella Errigo, Natasha A. Griffin, Jansen C. Howe, Emily L. Meadows, Elizabeth Kujanpaa, Monterey L. Nelson, Elysse Ostlund, Nicholas J. Suiter, Kaylee Tanner, Joseph Tolworthy, Maria Camila Vargas, Benjamin W. Abbott.

**Project administration:** Erin Fleming Jones, Rebecca J. Frei, Rhetta Shoemaker, Rachel Watts, Benjamin W. Abbott.

**Resources:** Zachary T. Aanderud, Neil C. Hansen, Benjamin W. Abbott.

**Supervision:** Erin Fleming Jones, Zachary T. Aanderud, Neil C. Hansen, Benjamin W. Abbott.

**Validation:** Erin Fleming Jones, Rebecca J. Frei, Raymond M. Lee, Benjamin W. Abbott.

**Writing – original draft:** Erin Fleming Jones, Rebecca J. Frei, Jordan D. Maxwell, Rhetta Shoemaker, Andrew P. Follett, Gabriella M. Lawson, Madeleine Malmfeldt, Rachel Watts, Zachary T. Aanderud, Carter Allred, Madeline Buhman, Hunter Burbidge, Amber Call, Isabella Errigo, Jansen C. Howe, Adam J. Norris, Nicholas J. Suiter, Joseph Tolworthy, Benjamin W. Abbott.

**Writing – review & editing:** Erin Fleming Jones, Rebecca J. Frei, Raymond M. Lee, Andrew P. Follett, Gabriella M. Lawson, Madeleine Malmfeldt, Rachel Watts, Zachary T. Aanderud, Carter Allred, Allison Tuttle Asay, Madeline Buhman, Amber Call, Natasha A. Griffin, Neil C. Hansen, Adam J. Norris, Nicholas J. Suiter, Benjamin W. Abbott.

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
