## [Decision Letter · Decision Letter 0]

13 May 2021

PONE-D-21-06637

Citizen science reveals unexpected solute patterns in semiarid river networks

PLOS ONE

Dear Dr. Fleming Jones,

Thank you for submitting your manuscript to PLOS ONE. After careful consideration, we feel that it has merit but does not fully meet PLOS ONE’s publication criteria as it currently stands. Therefore, we invite you to submit a revised version of the manuscript that addresses the points raised during the review process.

Thank you for an interesting contribution to watershed eutrophication, using citizen science. As you can see, Reviewer #2 recommended immediate acceptance of your manuscript.

However, Reviewer #1 raised some scepticism regarding using citizen science for this type of study, and I believe that addressing this Reviewer's specific comments, particularly in the Methods and Discussion sections, could help you to exemplify pros and cons with citizen science in a more informed way.

The final sentence of the first review suggested additional improvements to the Discussion, unfortunately without specifying which ones. After having read your Discussion section carefully again, I lack suggestions myself on what else to improve.

Please make sure to address all comments from me and each Reviewer in your rebuttal letter.

We look forward to receiving your revised manuscript.

Kind regards,

Andreas C. Bryhn

Academic Editor

PLOS ONE

Journal Requirements:

2. Please provide additional details regarding participant consent. In the ethics statement in the Methods and online submission information, please ensure that you have specified (1) whether consent was informed and (2) what type you obtained (for instance, written or verbal, and if verbal, how it was documented and witnessed). If the need for consent was waived by the ethics committee, please include this information.

3. In your Methods section, please provide additional information about the participant recruitment method and the demographic details of your participants. Please ensure you have provided sufficient details to replicate the analyses such as: a) the recruitment date range (month and year), b) a description of any inclusion/exclusion criteria that were applied to participant recruitment, c) a table of relevant demographic details, d) a statement as to whether your sample can be considered representative of a larger population, e) a description of how participants were recruited, and f) descriptions of where participants were recruited and where the research took place.

4. In your Methods section, please provide additional location information of the sampling sites, including geographic coordinates for the data set if available.

5. In your Methods section, please provide additional information regarding the permits you obtained for the work. Please ensure you have included the full name of the authority that approved the field site access and, if no permits were required, a brief statement explaining why.

6. We note that Figures 1 and S1Fig in your submission contain map/satellite images which may be copyrighted. All PLOS content is published under the Creative Commons Attribution License (CC BY 4.0), which means that the manuscript, images, and Supporting Information files will be freely available online, and any third party is permitted to access, download, copy, distribute, and use these materials in any way, even commercially, with proper attribution. For these reasons, we cannot publish previously copyrighted maps or satellite images created using proprietary data, such as Google software (Google Maps, Street View, and Earth). For more information, see our copyright guidelines: http://journals.plos.org/plosone/s/licenses-and-copyright.

6.1.    You may seek permission from the original copyright holder of Figures 1 and S1Fig to publish the content specifically under the CC BY 4.0 license. 

6.2.    If you are unable to obtain permission from the original copyright holder to publish these figures under the CC BY 4.0 license or if the copyright holder’s requirements are incompatible with the CC BY 4.0 license, please either i) remove the figure or ii) supply a replacement figure that complies with the CC BY 4.0 license. Please check copyright information on all replacement figures and update the figure caption with source information. If applicable, please specify in the figure caption text when a figure is similar but not identical to the original image and is therefore for illustrative purposes only.

7. We note that Figure S2Fig includes an image of a participant in the study. 

Reviewers' comments:

Reviewer's Responses to Questions

**Comments to the Author**

1. Is the manuscript technically sound, and do the data support the conclusions?

Reviewer #1: Partly

Reviewer #2: Yes

2. Has the statistical analysis been performed appropriately and rigorously? 

Reviewer #1: Yes

Reviewer #2: Yes

3. Have the authors made all data underlying the findings in their manuscript fully available?

Reviewer #1: No

Reviewer #2: Yes

4. Is the manuscript presented in an intelligible fashion and written in standard English?

Reviewer #1: Yes

Reviewer #2: Yes

5. Review Comments to the Author

Reviewer #1: This work collected samples from ~200 locations throughout the Utah Lake watershed on a single day in the spring, summer, and fall of 2018 by local volunteers. Spatial and seasonal dynamics of water quality indexes were addressed. Critical source area behavior for carbon, nitrogen, and phosphorus species was identified. Spatial variability of water quality indexes over watershed sizes was discussed. This study highlights that participatory science has great potential to reveal ecohydrological patterns and rehabilitate individual and community relationships with local ecosystems. Different with many previous studies, this study hired local volunteers to accomplish sampling works to save money and reduce temporal variability. However, was it efficient to catch temporal variability by once sampling event per season? For volunteers, did they really can make a standard field sampling work? For example, how did they get the water samples from the stream water? Did they collect water samples from several points to get a mixture sample for each sampling site? Importantly, what did all volunteers response to the sampling work? Will the participate into water quality protection, supervision and management due to this sampling work? Authors proposed an hourglass shape pattern of spatial variability over watershed size and suggested that this was attributable to the distribution of human activity and hydrological complexity associated with return flows, losing river reaches, and diversions in the tailwaters. However, it is lack of evidences to support, e.g., hydrological data and pollution source data. In addition, discussions should be substantially improved to address mechanisms and implications.

Reviewer #2: I carefully reviewed the manuscript "Citizen science reveals unexpected solute patterns in semiarid river networks". The text is well written and brings relevant information that relate aspects related to the monitoring of semi-arid rivers and the participation of the community inserted in the hydrographic basins.

The methodology was properly used and the analytical techniques are already part of consolidated protocols. The results are presented in a straightforward way and are adequately discussed. I therefore consider the manuscript approved for publication.

6. PLOS authors have the option to publish the peer review history of their article (what does this mean?). If published, this will include your full peer review and any attached files.

Reviewer #1: No

Reviewer #2: **Yes: **Francisco José de Paula Filho

---

## [Author Response · Author response to Decision Letter 0]

7 Jul 2021

Dear Dr. Bryhn,

We greatly appreciate the speed and thoroughness of your handling of this manuscript. We have carefully revised the paper based on your guidance and the reviewer criticisms. We provide detailed descriptions of the changes and responses to the input below (original input from you or reviewers in blue, our responses in black). Specifically, we now provide more justification for the methods (e.g., the participatory science approach, sampling frequency, spatial extent, etc.) and try to more fully integrate this work into the diverse and rapidly evolving literature concerning water quality and aquatic ecosystems in the Anthropocene. We would be happy to provide additional information if needed.

Thank you again and best wishes,

Erin Jones, on behalf of all co-authors

Dear Dr. Fleming Jones,

Thank you for submitting your manuscript to PLOS ONE. After careful consideration, we feel that it has merit but does not fully meet PLOS ONE’s publication criteria as it currently stands. Therefore, we invite you to submit a revised version of the manuscript that addresses the points raised during the review process.

Thank you for an interesting contribution to watershed eutrophication, using citizen science. As you can see, Reviewer #2 recommended immediate acceptance of your manuscript.

However, Reviewer #1 raised some scepticism regarding using citizen science for this type of study, and I believe that addressing this Reviewer's specific comments, particularly in the Methods and Discussion sections, could help you to exemplify pros and cons with citizen science in a more informed way.

The final sentence of the first review suggested additional improvements to the Discussion, unfortunately without specifying which ones. After having read your Discussion section carefully again, I lack suggestions myself on what else to improve.

Please make sure to address all comments from me and each Reviewer in your rebuttal letter.

Thank you for these useful comments and criticisms. We have revised the methods and discussion as suggested to demonstrate how widespread this methodology is and to communicate the strengths and limitations of participatory science. We have also performed a final proof reading to update citations throughout and tighten the text.

We have reviewed file names and all follow PLOS ONE’s conventions.

2. Please provide additional details regarding participant consent. In the ethics statement in the Methods and online submission information, please ensure that you have specified (1) whether consent was informed and (2) what type you obtained (for instance, written or verbal, and if verbal, how it was documented and witnessed). If the need for consent was waived by the ethics committee, please include this information.

When we were planning the project, we consulted the Institutional Review Board (IRB) at Brigham Young University (BYU), which oversees all research involving human subjects. Because participants were not the subject of the research (i.e., we did not collect information about their identities or experiences), we were informed that IRB approval was not needed. We now specify this in the methods. 

3. In your Methods section, please provide additional information about the participant recruitment method and the demographic details of your participants. Please ensure you have provided sufficient details to replicate the analyses such as: a) the recruitment date range (month and year), b) a description of any inclusion/exclusion criteria that were applied to participant recruitment, c) a table of relevant demographic details, d) a statement as to whether your sample can be considered representative of a larger population, e) a description of how participants were recruited, and f) descriptions of where participants were recruited and where the research took place.

We now provide greater detail on how we recruited community members in the methods section, including the duration of the recruitment period, what channels were used, and the fact that there were no inclusion/exclusion criteria. Following the guidance of the IRB, we did not collect demographic information on participants, and we are not able to draw conclusions about representativeness of the participating population. We recognize this is a limitation and plan to address it in future work. 

4. In your Methods section, please provide additional location information of the sampling sites, including geographic coordinates for the data set if available.

The coordinates are included in the linked database hosted by a permanent data repository (HydroShare). We have updated the text to ensure full accessibility.

5. In your Methods section, please provide additional information regarding the permits you obtained for the work. Please ensure you have included the full name of the authority that approved the field site access and, if no permits were required, a brief statement explaining why.

We have updated the methods to include this (line 178). 

6. We note that Figures 1 and S1 Fig in your submission contain map/satellite images which may be copyrighted. All PLOS content is published under the Creative Commons Attribution License (CC BY 4.0), which means that the manuscript, images, and Supporting Information files will be freely available online, and any third party is permitted to access, download, copy, distribute, and use these materials in any way, even commercially, with proper attribution. For these reasons, we cannot publish previously copyrighted maps or satellite images created using proprietary data, such as Google software (Google Maps, Street View, and Earth). For more information, see our copyright guidelines: http://journals.plos.org/plosone/s/licenses-and-copyright.

 All of the included maps are fully compatible with PLOS ONE’s CC BY 4.0 license. We have attached proof of permissions and updated the captions accordingly.

7. We note that Figure S2Fig includes an image of a participant in the study. 

Figure S2 contains an image of a co-author (Gabriella M Lawson), who was demonstrating the sampling technique in the flyer we used to train volunteers. We obtained her permission to use the photo before printing the training flyers.

Reviewer #1: This work collected samples from ~200 locations throughout the Utah Lake watershed on a single day in the spring, summer, and fall of 2018 by local volunteers. Spatial and seasonal dynamics of water quality indexes were addressed. Critical source area behavior for carbon, nitrogen, and phosphorus species was identified. Spatial variability of water quality indexes over watershed sizes was discussed. This study highlights that participatory science has great potential to reveal ecohydrological patterns and rehabilitate individual and community relationships with local ecosystems. Different with many previous studies, this study hired local volunteers to accomplish sampling works to save money and reduce temporal variability. However, was it efficient to catch temporal variability by once sampling event per season? For volunteers, did they really can make a standard field sampling work? For example, how did they get the water samples from the stream water? Did they collect water samples from several points to get a mixture sample for each sampling site? Importantly, what did all volunteers response to the sampling work? Will the participate into water quality protection, supervision and management due to this sampling work? Authors proposed an hourglass shape pattern of spatial variability over watershed size and suggested that this was attributable to the distribution of human activity and hydrological complexity associated with return flows, losing river reaches, and diversions in the tailwaters. However, it is lack of evidences to support, e.g., hydrological data and pollution source data. In addition, discussions should be substantially improved to address mechanisms and implications.

We thank the reviewer for their careful read and criticisms of the manuscript. We make one correction before addressing the critiques: participants were not hired (i.e., paid or compensated), they volunteered as a part of an open community event. We mention this in the revised manuscript.

Concerning the questions about sampling design, we address each in order below:

was it efficient to catch temporal variability by once sampling event per season? 

For our sampling design, there was a fundamental tradeoff between frequency and extent. We could either sample a few locations at higher frequency or sample more locations at lower frequency. Informed by recent research in temperate and Arctic environments that indicated relatively persistent spatial patterns in network-scale water chemistry (Abbott et al. 2018a; Dupas et al. 2019; Shogren et al. 2019), we chose to favor the spatial extent of sampling in this study. There is surely important temporal variation that our study does not capture, but our focus provides a relatively unique view by capturing near synchronous chemical conditions across a large semi-arid watershed.

For volunteers, did they really can make a standard field sampling work? For example, how did they get the water samples from the stream water? Did they collect water samples from several points to get a mixture sample for each sampling site? 

With all participatory science, there is the risk of having less rigorous and controlled conditions than traditional laboratory and field conditions carried out by professional researchers (Breuer et al. 2015; Abbott et al. 2018b). There are common techniques to mitigate this tradeoff, including careful training, simplification of procedures, and replication of sampling (i.e., having multiple volunteers complete the same task and compare the result). We used all of these techniques, providing training on the day of each sampling, using an extremely simple sample collection and preservation technique, and assigning a small percentage of volunteers to resample duplicate locations. Because of the simplified procedure, volunteers did not collect a composite or “mixture” sample; they filled a single syringe from the bank of the stream or lake. The great benefit of involving nonprofessional volunteers is that it expands the scope of potential sample collection. The large number of samples, along with the precautions described above, provides a robust approach to reducing error associated with sampling and transport. We have expanded our justification of this approach in the methods section.

Importantly, what did all volunteers response to the sampling work? Will the participate into water quality protection, supervision and management due to this sampling work?

We share the reviewer’s interest in the quality and effect of the event on the perceptions of the participants. We plan to gather this information during future events, but unfortunately, we do not have quantitative information to this effect for the current manuscript. We do have quite a bit of qualitative information, which we now try to incorporate in the discussion. One useful metric for assessing impact is that we have received a large number of requests to continue the project from past participants and partner organizations who contributed to the effort.

Reviewer #2: I carefully reviewed the manuscript "Citizen science reveals unexpected solute patterns in semiarid river networks". The text is well written and brings relevant information that relate aspects related to the monitoring of semi-arid rivers and the participation of the community inserted in the hydrographic basins.

The methodology was properly used and the analytical techniques are already part of consolidated protocols. The results are presented in a straightforward way and are adequately discussed. I therefore consider the manuscript approved for publication.

We thank the reviewer for their careful read of the manuscript and positive assessment.

---

## [Editor Report · Decision Letter 1]

16 Jul 2021

Citizen science reveals unexpected solute patterns in semiarid river networks

PONE-D-21-06637R1

Dear Dr. Jones,

We’re pleased to inform you that your manuscript has been judged scientifically suitable for publication and will be formally accepted for publication once it meets all outstanding technical requirements.

Kind regards,

Andreas C. Bryhn

Academic Editor

PLOS ONE
---

## [Editor Report · Acceptance letter]

11 Aug 2021

PONE-D-21-06637R1 

Citizen science reveals unexpected solute patterns in semiarid river networks 

Dear Dr. Jones:

I'm pleased to inform you that your manuscript has been deemed suitable for publication in PLOS ONE. Congratulations! Your manuscript is now with our production department. 

Kind regards, 

on behalf of

Dr. Andreas C. Bryhn 

Academic Editor

PLOS ONE